REGISTERED REPORT PROTOCOL

# Protocol for the systematic review of the *Pneumocystis jirovecii*-associated pneumonia in non-HIV immunocompromised patients

**Mauricio Ernesto Orozco-Ugarriza**[1,2]*, **Yenifer Olivo-Martínez**[1,3], **Yuranis E. Rodger-Cervantes**[4]

1 Grupo de Investigación en Microbiología y Ambiente (GIMA), Universidad de San Buenaventura, Cartagena, Colombia, 2 Grupo de Investigación Traslacional en Biomedicina y Biotecnología (GITB&B), Corporación para el Desarrollo de la Investigación en Biomedicina & Biotecnología, Cartagena, Colombia, 3 Biochemistry and Diseases Research Group, Facultad de Medicina, Universidad de Cartagena, Cartagena, Colombia, 4 Graduated from the Bacteriology and Clinical Laboratory Program, Faculty of Health Sciences, Universidad de San Buenaventura Cartagena, Cartagena, Colombia

☙ These authors contributed equally to this work.
* morozcou@usbcartagena.edu.co, mauricioorozcou@gmail.com

This is a Registered Report and may have an associated publication; please check the article page on the journal site for any related articles.

## Abstract

### Introduction

*Pneumocystis jirovecii* pneumonia (PJP) is a well-known and frequent opportunistic infection in HIV patients. However, there has been an increase in the number of reports of PJP in other immunosuppressed patients with autoimmune inflammatory disorders or because of chemotherapy and high doses of steroids, especially when used in combination as part of immunosuppressive therapy.

### Objective

Despite the increasing importance of PJP in non-HIV patients, there is a lack of comprehensive and updated information on the epidemiology, pathogenesis, diagnosis, microbiology, treatments, and prophylaxis of this infection in this population. Therefore, the objective of this systematic review is to synthesize information on these aspects, from a perspective of evidence-based medicine.

### Methods

The protocol is prepared following the preferred reporting items for systematic reviews and meta-analyses (PRISMA-P) guidelines. We will perform a systematic review of literature published between January 2010 and July 2023, using the databases PubMed, Google Scholar, ScienceDirect, and Web of Science. In addition, manual searches will be carried out through related articles, and references to included articles. The main findings and clinical outcomes were extracted from all the eligible studies with a standardized instrument. Two authors will independently screen titles and abstracts, review full texts, and collect data. Disagreements will be resolved by discussion, and a third reviewer will decide if there

**Data Availability Statement:** No datasets were generated or analysed during the current study. All relevant data from this study will be made available upon study completion.

**Funding:** The author(s) received no specific funding for this work.

**Competing interests:** The authors declare that they have no competing interests.

is no consensus. We will synthesize the results using a narrative or a meta-analytic approach, depending on the heterogeneity of the studies.

## Expected results

It is expected that this systematic review will provide a comprehensive and up-to-date overview of the state-of-the-art of PJP in non-HIV patients. Furthermore, the study will highlight possible gaps in knowledge that should be addressed through new research.

## Conclusions

Here, we present the protocol for a systematic review which will consider all existing evidence from peer-reviewed publication sources relevant to the primary and secondary outcomes related to diagnosing and managing PJP in non-HIV patients.

## Introduction

*Pneumocystis jirovecii* pneumonia (PJP) is a fungal infection that mainly affects patients with immune system disorders. It is a severe and potentially fatal disease, especially in immunocompromised patients. One of the most common causes of immunosuppression is human immunodeficiency virus (HIV) infection, which is strongly associated with PJP [1]. However, in recent years, the incidence of PJP has decreased in HIV patients is around 10–20% [2], due to antiretroviral therapy and routine prophylaxis with trimethoprim-sulfamethoxazole (TMP-SMX) [3].

In contrast, the prevalence of PJP has risen steadily and has emerged as a major cause of morbidity and mortality among patients with non-HIV immunosuppression [4]. These patients have different causes of immune system alterations that can increase the risk of developing PJP, such as hematological or solid malignancies [5], organ transplantation [2], inflammatory or autoimmune diseases [6], as well as immunosuppressive treatments [6, 7].

The clinical presentation of PJP may differ between patients with and without HIV infection [8], which may influence early detection and initiation of appropriate treatment [9]. Studies have shown that PJP in non-HIV patients experiences a more progressive clinical course, a poor prognosis, and a higher mortality rate (ranging from 30% to 60%) than in HIV patients [4, 10]. This is due to several factors, such as delay in diagnosis, lack of prevention, advanced age, comorbidities, and immunosuppression. Therefore, early diagnosis is key to allowing evaluation of the patient's clinical status and choice of the best anti-PJP therapy, which may include TMP-SMX or other options, as appropriate, to improve prognosis [11].

The prevention and management of PJP in non-HIV patients remain a challenge in the medical field. There is currently no standard guideline for the prophylaxis of this infection in this group, and the decision to start prophylaxis depends on the individual risk-benefit assessment for each patient [4]. It is essential to improve our understanding of the risk factors and to carry out a personalized evaluation of the patients [12]. An early diagnosis and appropriate therapy can make a difference in the prognosis of the disease [13]. Several studies have identified risk factors for PJP in non-HIV patients, such as the concomitant use of corticosteroids and immunosuppressants, lymphopenia, and a CD4+ T cell count below 200 cells/μL [14, 15]. However, it is crucial to highlight that the CD4+ T cell count, used as an indicator of immune function and the risk of developing opportunistic infections in HIV patients, is not a reliable criterion for the prophylaxis of PJP in non-HIV patients [4, 10]. This is because it can vary depending on the cause of immunosuppression and not reflect the actual immune status of the patient [16].

The diagnosis of PJP is based on clinical presentation, radiological findings, and confirmation by observing the organisms in stained respiratory samples [17]. However, *P. jirovecii* is extremely difficult to cultivate *in vitro*, which poses significant challenges in the diagnosis, treatment, and prevention of the disease. Despite these difficulties, there are currently more efficient laboratory tools for the diagnosis of *P. jirovecii*, such as polymerase chain reaction (PCR) and immunofluorescence assays, which have improved our ability to identify cases of PJP. These methods offer increased sensitivity and a faster turnaround time, which are critical in clinical settings [13, 18].

However, PCR's high sensitivity also presents the challenge of differentiating between actual infection and colonization since it can detect low levels of organisms that may not necessarily signify a clinically relevant active infection, thereby increasing the probability of false positive assays. Moreover, the assay's sensitivity depends on gene target selection. Consequently, a careful interpretation of PCR results in the context of clinical presentation and other diagnostic findings is necessary [19, 20].

In a recent 2020 study, Liu et al. reported a MultiCode real-time PCR assay targeting the mitochondrial small subunit (mtSSU) rRNA gene, designed to detect *P. jirovecii* in Bronchoalveolar lavage fluid (BALF) samples, achieving results in a reduced time of 2.5 hours. In comparison with reference methods such as the Pneumocystis-specific direct fluorescent antibody assay (DFA) and the mtLSU gene-targeting PCR assay, this assay was able to detect a rate of 800 copies/mL, translating to 22 organisms/mL, with a sensitivity of 96.9% and a specificity of 94.6%. The study proposes specific threshold cycle cut-off value (CT) criteria for the mtSSU gene to clinically interpret the results and distinguish between colonization and active infection. An observed CT value of ≤ 39.1 suggests a probable or confirmed diagnosis of PJP, indicating a significant pathogen load and a high probability of active infection. On the other hand, a high CT in the range of 40.1 to 45.0 is considered a non-significant amplification result, indicating a low concentration of *P. jirovecii* DNA, which could suggest colonization or the presence of residual DNA without a clinically relevant active infection [21].

Despite advances in diagnostic techniques for PJP [13, 18], the difficulty of cultivating this organism under *in vitro* conditions persists. This situation can lead to delays in confirming the presence of the organism in clinical samples, which in turn can delay the initiation of effective therapies [22]. Furthermore, it limits the ability to conduct detailed research on the biological characteristics of the pathogen and its response to treatments, including the host immune responses to the organism and the immunopathology of this infection [23]. Given this limitation, the current information on the basic aspects of PJP has been obtained mainly from some limited clinical studies and experiments in rodent models [24]. These sources of information may not fully reflect the complexity of PJP in human patients, so more research is required to improve the diagnosis, prevention, and treatment of this infection [25].

Considering the serious consequences of PJP in immunocompromised patients due to causes other than HIV, in this article, we present a protocol to perform a systematic review. This systematic review aims to synthesize information on the epidemiology, pathogenesis, diagnosis, microbiology, treatments, and prophylaxis of PJP in these patients, from an evidence-based medicine perspective. This systematic review is expected to provide clarity on the current state of knowledge and identify research gaps, ultimately contributing to better management of PJP in non-HIV patients.

## Methods

### Design and registration

This systematic review protocol has been conducted following the recommendations of the Preferred Reporting Items for Systematic Reviews and Meta-Analysis (PRISMA-P) guidelines

[26]. This review protocol was registered in the Prospective Register of Systematic Reviews platform (PROSPERO) [27], (registration number CRD42023429112), and can be accessed at https://www.crd.york.ac.uk/prospero/display_record.php?ID=CRD42023429112. See (S1 Checklist) for the completed PRISMA-P checklist. We will only make changes to the study protocol if necessary and we will report them in the PROSPERO record.

## Search strategy

The search strategy will be conducted comprehensively across electronic databases, including PubMed, Google Scholar, ScienceDirect, and Web of Science. Controlled terms (also known as Medical Subject Headings or MeSH terms) and relevant keywords will be used to identify relevant studies related to *Pneumocystis jirovecii*-associated pneumonia in non-HIV immunosuppression, such as "*Pneumocystis jirovecii*", "*Pneumocystis carinii*", "pneumocystosis", "pneumonia", "non-HIV", "without HIV diagnosis", "immunocompromised", "immunosuppression", "cancer", "transplantation", "autoimmune", and "inflammatory", "incidence", "clinical characteristics", "risk factors" and "therapeutic management".

We will use Boolean operators (such as AND, OR, NOT) and filters to refine the search results. We will also perform manual searches through related articles, and references of included articles, to identify additional relevant studies.

The pilot search will be adapted to each database, avoiding any inconsistencies that may affect data extraction and ensuring that the final search strategy is correctly adapted. The last pilot search was conducted in October 2023. The details of the search strategy, including the terms used and filters applied, will be recorded to ensure transparency and reproducibility of the systematic review. The details of a search strategy sample conducted on PubMed in October 2023 can be found in the supporting information (S1 File).

## Eligibility criteria

We will include studies that meet the following criteria:

**Population.**   Patients with non-HIV immunosuppression, defined as having any condition or treatment that alters the immune system function, such as hematological or solid malignancies, organ transplantation, inflammatory or autoimmune diseases, or immunosuppressive drugs.

**Intervention.**   Any intervention for the diagnostic, therapeutic, or prophylactic measure for PJP, such as laboratory tests; radiological exams; antifungal drugs; and preventive measures. The type, duration, frequency, dosage, route, or any other relevant characteristic of the measure will be specified, respectively.

**Exposure.**   Any risk factor, cause, agent, or situation associated with PJP, such as immunosuppression, organ transplantation, autoimmune diseases, corticosteroid use, exposure to the *P. jirovecii fungus*, or underlying medical conditions, immunosuppressive therapies, environmental exposures, or genetic predisposition, will be specified for type, duration, frequency, dosage, route, or any other relevant characteristic.

**Comparator.**   We will have two types of comparators including 1) HIV patients with PJP, the most well-known and common group affected by this infection. 2) Any comparator for the intervention, such as different tests, drugs, doses, durations, or strategies.

**Outcome.**   Any outcome related to PJP, such as incidence, prevalence, mortality, morbidity, and adverse effects.

**Study design.**   We will include original articles with an observational and interventional design, as well as systematic reviews and meta-analyses, that report data on PJP in non-HIV patients and outcomes of interest (incidence, clinical characteristics, risk factors, and

therapeutic management). We will exclude case reports, case series, letters, editorials, and reviews that do not provide original data.

**Publication date.**  The search will be limited to studies published between January 2010 and July 2023. In addition, the search period may be extended until December 2023, aiming to retrieve the most up-to-date evidence on PJP in non-HIV patients. This will be achieved through manual searches, reviewing the bibliographic references of included articles, and considering expert recommendations.

**Language.**  We will include studies published in English or Spanish, as these are the languages that the reviewers can read and understand. However, efforts will be made to identify and access relevant studies published in other languages if possible.

## Literature screening and study selection

We plan to make use of a flow diagram to summarize the article screening process and detail the reasons for the exclusion of studies screened as full text. This will follow the PRISMA guidelines for reporting systematic reviews [28]. The report will feature a PRISMA-compliant flow chart illustrating the process of study screening and selection, as depicted in the S2 File.

All studies retrieved by the searches will be combined using the Mendeley software desktop, and duplicates will be removed. All articles returned by the search procedure will be divided between the authors (MEOU and YRC), who will independently conduct a 3-step screening (titles, abstracts, and full texts) and decide on their inclusion or exclusion. Any disagreement between the reviewers will be resolved by discussion, and a third reviewer (YOM) will decide if there is no consensus.

## Data extraction

Data will be collected independently by three individual reviewers from each eligible publication, and the extracted data items, using a standardized data extraction form (See S3 File). The data will be exported to Excel (the spreadsheet software). After completing the selection and screening of the studies, data extraction will be conducted in four stages: 1) development of a data extraction form, 2) peer-review of the data extraction form, 3) pilot test of the data extraction form, and 4) final data extraction.

## Data items

The following data items will be extracted from the included studies:

**Identification of the study.**  This will include the name of the journal, article DOI, article title, authors, publication year, short citation, and country.

**Methods.**  This will include the study objectives, study design, inclusion and exclusion criteria, intervention (type, dose, duration, frequency, mode of administration), intervention characteristics, comparator characteristics (if applicable), diagnostic methods, participant demographics (mean age, sex, number of participants), and description of effect and effect size (if applicable), and results (summary statistics, effect estimates, confidence intervals, p-values, subgroup analyses, sensitivity analyses, risk of bias, quality of evidence).

**Main findings.**  This will include diagnosis, patient characteristics, and other relevant clinical outcome measures (definition, assessment, time point). One-sentence conclusion of the study.

## Methodological quality assessment

For the systematic review, the method of assessing the risk of bias or study quality will involve the following steps:

**Independent assessment.** Two reviewers (MEOU and YRC), will independently evaluate the methodological quality of the included studies or literature.

**Disagreement resolution.** Any disagreements between reviewers' assessments will be resolved through discussion and consensus, in cases where consensus cannot be reached, a third reviewer (YOM) will be involved to reach a final agreement.

**Study selection and data extraction.** Two reviewers will independently assess the search results and select high-quality literature for inclusion in the review. Multiple reviewers will be involved in the data extraction process, to minimize the risk of errors.

**Blinding.** The study selection, data extraction, and risk of bias assessment will be performed without blinding the assessors to the study authors or the journal of publication.

These measures are implemented to ensure the rigor and reliability of the review process. By involving multiple independent reviewers and resolving disagreements through discussion and consensus, the risk of bias in the included studies will be assessed transparently and comprehensively.

## Strategy for data synthesis

We plan to synthesize the data using a systematic narrative synthesis approach, to provide a comprehensive view of the topic by presenting a coherent narrative that integrates the primary findings from the included studies. This synthesis will involve summarizing data extracted from the included studies, with a specific focus on clinical characteristics, risk factors, therapeutic treatments, and other interventions, as well as the methodological quality of the studies and any identified limitations. When appropriate, we will categorize these findings into relevant groups to facilitate a clearer presentation of the synthesized data. Additionally, we will employ descriptive statistics to analyze the results, where feasible. These measures will contribute to a comprehensive and coherent data synthesis, enabling a deeper understanding of the topic, its clinical implications, and the quality of the underlying evidence.

## Ethics

This is a systematic review that will use published data and does not require ethical approval.

## Status of the study and dissemination plan

This systematic review is currently in the pilot and preliminary search phase, awaiting approval of the Registered Report Protocol. We expect to complete the project, publish the results in a peer-reviewed journal, and report them within the next 12 months. We will follow the updated PRISMA guideline to report the final paper and register it on the PROSPERO website. All the data will be accessible as supplementary information when the review is completed. Moreover, we will also present the main findings at academic conferences and relevant professional meetings.

## Discussion

*Pneumocystis jirovecii* pneumonia (PJP) is an opportunistic infection that primarily affects patients with immune system disorders, such as those with human immunodeficiency virus (HIV) [1]. However, in recent years, the incidence of PJP in developing countries has decreased by around 10–20% among HIV patients [2], due to antiretroviral therapy and routine prophylaxis with trimethoprim-sulfamethoxazole (TMP-SMX) [3], while other AIDS-associated opportunistic infections such as tuberculosis and enteric pathogens have become predominant. Moreover, more recent studies have demonstrated a high seroprevalence of *P. jirovecii*, in healthy individuals with HIV as well as elevated rates of clinical disease among

patients with non-HIV immunosuppression, including those undergoing corticosteroid treatment, chemotherapy, organ transplants, or managing autoimmune diseases [4–7]. Apparently, from an epidemiologic viewpoint, the increase in prevalence among patients with non-HIV immunosuppression may be explained possibly as a result of improved access to diagnostics and treatment [8].

However, in the clinical context, there is a lack of consensus on several aspects of PJP management, related to the criteria for indicating prophylaxis, microbiological diagnosis, choice of treatment, and duration [9]. The lack of consensus is partly attributed to differences in the clinical presentation of PJP between patients based on the presence or absence of HIV infection, which can affect the quality of care patients receive, leading to several important implications, such as inaccurate diagnosis, a poor prognosis, and a higher mortality rate (ranging from 30% to 60%) compared to patients with HIV [4, 10]. A better understanding of these factors would directly support clinical decision-making, leading to improved outcomes. By achieving an improved, standardized, and effective PJP management for all patients, regardless of HIV status.

Considering the serious consequences of PJP in immunocompromised patients due to causes other than HIV, in this systematic review protocol, we describe a detailed plan of all steps to develop a review of Pneumocystis jirovecii pneumonia (PJP) in non-HIV immunocompromised patients, including protocol development, search strategy, selection. of studies, data extraction, quality assessment, and data synthesis. By rigorously following this protocol, we aim to generate unbiased evidence on the clinical characteristics, risk factors, treatments, and available interventions for this illness.

The knowledge obtained from this systematic review can contribute to better clinical practice and future research on Pneumocystis jirovecii pneumonia in non-HIV patients. On the one hand, we expect that our review will provide a comprehensive and updated overview of the state of the art on the epidemiology, pathogenesis, diagnosis, microbiology, treatments, and prophylaxis of this infection in this population.

Finally, we hope to identify existing knowledge gaps that may form the basis of possible future lines of research on various aspects of Pneumocystis jirovecii pneumonia (PJP) in non-HIV immunocompromised patients, that may contribute to improving the prevention, diagnosis, and management of this disease.

## Conclusion

The PJP is a serious and potentially fatal opportunistic infection that affects immunocompromised patients, especially those who are not infected by HIV. The diagnosis and management of PJP in non-HIV patients pose significant challenges, as there are no clear guidelines or consensus on the optimal strategies. Therefore, it is important to conduct a systematic review that synthesizes the available evidence on the clinical characteristics, risk factors, treatments, and interventions for PJP in non-HIV patients. In this protocol, we describe the methods and criteria that we will use to perform a comprehensive and up-to-date search, selection, extraction, quality assessment, and synthesis of data from peer-reviewed publications relevant to the primary and secondary outcomes of interest. We expect that this systematic review will provide a state-of-the-art overview of the current knowledge and practice of PJP in non-HIV patients, as well as identify the existing knowledge gaps and suggest possible future lines of research.

## Supporting information

**S1 Checklist. Preferred Reporting Items for Systematic Review and Meta-Analysis Protocols (PRISMA-P checklist).**
(DOCX)

**S1 File. Sample search strategy for PubMed.**
(DOCX)

**S2 File. PRISMA 2009 flow diagram.**
(DOC)

**S3 File. Data extraction form for systematic review.**
(DOCX)

**S4 File. PROSPERO registration.**
(PDF)

## Acknowledgments

The authors would like to thank all members of the group for their support and help.

## Author Contributions

**Conceptualization:** Mauricio Ernesto Orozco-Ugarriza, Yenifer Olivo-Martínez, Yuranis E. Rodger-Cervantes.

**Data curation:** Mauricio Ernesto Orozco-Ugarriza, Yenifer Olivo-Martínez, Yuranis E. Rodger-Cervantes.

**Investigation:** Yuranis E. Rodger-Cervantes.

**Methodology:** Mauricio Ernesto Orozco-Ugarriza, Yuranis E. Rodger-Cervantes.

**Supervision:** Mauricio Ernesto Orozco-Ugarriza.

**Validation:** Mauricio Ernesto Orozco-Ugarriza, Yenifer Olivo-Martínez, Yuranis E. Rodger-Cervantes.

**Visualization:** Mauricio Ernesto Orozco-Ugarriza, Yenifer Olivo-Martínez, Yuranis E. Rodger-Cervantes.

**Writing – original draft:** Mauricio Ernesto Orozco-Ugarriza, Yenifer Olivo-Martínez, Yuranis E. Rodger-Cervantes.

**Writing – review & editing:** Mauricio Ernesto Orozco-Ugarriza, Yenifer Olivo-Martínez, Yuranis E. Rodger-Cervantes.

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
