## [Decision Letter · Decision Letter 0]

10 Jan 2024

PONE-D-23-38915Protocol for the systematic review of the Pneumocystis jirovecii-associated pneumonia in non-HIV immunocompromised patientsPLOS ONE

Dear Dr. Orozco Ugarriza,

Thank you for submitting your manuscript to PLOS ONE. After careful consideration, we feel that it has merit but does not fully meet PLOS ONE’s publication criteria as it currently stands. Therefore, we invite you to submit a revised version of the manuscript that addresses the points raised during the review process.

We look forward to receiving your revised manuscript.

Kind regards,

Benjamin M. Liu, MBBS, PhD, D(ABMM), MB(ASCP)

Academic Editor

PLOS ONE

Journal Requirements:

2. In your cover letter, please confirm that the research you have described in your manuscript, including participant recruitment, data collection, modification, or processing, has not started and will not start until after your paper has been accepted to the journal (assuming data need to be collected or participants recruited specifically for your study). In order to proceed with your submission, you must provide confirmation.

Additional Editor Comments:

Editor's comments

1. line 91-103: the authors failed to provide updated introduction on the novel molecular testing for PCP. Please cite the following reference and discuss the progress/advantages (e.g., increased sensitivity and fast turnaournd time, sample to answer detection) and challenges (difficulty in differentiating colocalization vs true infection) of PCR for PJP:

Liu B, Totten M, Nematollahi S, Datta K, Memon W, Marimuthu S, Wolf LA, Carroll KC, Zhang SX. Development and Evaluation of a Fully Automated Molecular Assay Targeting the Mitochondrial Small Subunit rRNA Gene for the Detection of Pneumocystis jirovecii in Bronchoalveolar Lavage Fluid Specimens. J Mol Diagn. 2020 Dec;22(12):1482-1493. doi: 10.1016/j.jmoldx.2020.10.003. Epub 2020 Oct 15. Erratum in: J Mol Diagn. 2021 Apr;23(4):506. PMID: 33069878.

2. The authors have written six paragraphs for Introduction. However, Discussion is just one paragraph, which is unacceptable. The authors are encouraged to discuss different aspects of PCP.

Comments from PLOS Editorial Office:

We noted that the decision includes recommendations that you cite specific previously published works. As always, we recommend that you please review and evaluate the requested works to determine whether they are relevant and should be cited. It is not a requirement to cite these works. We appreciate your attention to this request.

Reviewers' comments:

Reviewer's Responses to Questions

**Comments to the Author**

1. Does the manuscript provide a valid rationale for the proposed study, with clearly identified and justified research questions?

Reviewer #1: Yes

2. Is the protocol technically sound and planned in a manner that will lead to a meaningful outcome and allow testing the stated hypotheses?

Reviewer #1: Yes

3. Is the methodology feasible and described in sufficient detail to allow the work to be replicable?

Reviewer #1: Yes

4. Have the authors described where all data underlying the findings will be made available when the study is complete?

Reviewer #1: Yes

5. Is the manuscript presented in an intelligible fashion and written in standard English?

Reviewer #1: Yes

6. Review Comments to the Author

You may also provide optional suggestions and comments to authors that they might find helpful in planning their study.

Reviewer #1: A systematic review of the epidemiology, pathogen biology, pathogenesis, diagnosis and treatment, prevention, and other information of non-HIV PJP patients is of great practical significance.

This research protocol has been standardized, registered, and obtained a CRD number. We look forward to future research results.

Although this systematic review will be using published data, please confirm with the ethics committee of your faculty or university whether ethical approval is required, including the researcher's information, for instance. (Line 234)

7. PLOS authors have the option to publish the peer review history of their article (what does this mean?). If published, this will include your full peer review and any attached files.

Reviewer #1: No

---

## [Author Response · Author response to Decision Letter 0]

23 Mar 2024

We have carefully reviewed your suggestions and have also considered the reviewer's comments. Below, we present our responses to each point raised:

Line 91-103:

Response: We have updated the manuscript's introduction to include information on novel molecular testing for Pneumocystis jirovecii pneumonia (PCP). We have cited the provided reference and discussed the advances and challenges of PCR for PJP detection, including increased sensitivity, fast turnaround time, and challenges associated with differentiation between colocalization and true infection. Additionally, we have highlighted the work of Liu et al. (2020) and its impact in this field.

The number of paragraphs in the Discussion section:

Response: We have revised the manuscript's structure to ensure a broader and more detailed discussion of different aspects of PJP in the Discussion section. Our protocol follows the guidelines for PLOS ONE authors, emphasizing the strengths and limitations of the protocol, as well as the potential implications of the review. The Discussion section now encompasses multiple paragraphs exploring various aspects related to PJP, including clinical aspects, diagnosis, treatment, and future perspectives.

Line 234 - Reviewer's Comment:

We confirmed that this systematic review will utilize published data exclusively and does not involve any human subjects or data collection from individuals. Therefore, it does not necessitate ethical approval from the ethics committee of our faculty or university. We have clarified this point in the revised version of the manuscript.

---

## [Editor Report · Decision Letter 1]

27 Mar 2024

Protocol for the systematic review of the Pneumocystis jirovecii-associated pneumonia in non-HIV immunocompromised patients

PONE-D-23-38915R1

Dear Dr. Ugarriza,

We’re pleased to inform you that your manuscript has been judged scientifically suitable for publication and will be formally accepted for publication once it meets all outstanding technical requirements.

Kind regards,

Benjamin M. Liu, MBBS, PhD, D(ABMM), MB(ASCP)

Academic Editor

PLOS ONE

Additional Editor Comments (optional):

When performing proofreading, please change "Li" to "Liu" in line 104